# Numerical Simulation of Slag Entrainment by Vortex Flux during Tapping at Converter

**DOI:** 10.3390/ma16083209

**Published:** 2023-04-19

**Authors:** Chengyong Huang, Ye Sun, Wei Liu, Jingshe Li, Shufeng Yang, Jianfeng Dong

**Affiliations:** 1School of Metallurgical and Ecological Engineering, University of Science and Technology Beijing, Beijing 100083, China; b20200097@xs.ustb.edu.cn (C.H.);; 2Beijing Key Laboratory of Special Melting and Preparation of High-End Metal Materials, Beijing 100083, China

**Keywords:** converter, vortex, flow field analysis, numerical simulation

## Abstract

In order to improve the yield of steel produced in the converter and the quality of the molten steel, and to understand the distribution of the flow field in the converter and ladle during the steelmaking process, the CFD fluid simulation software Fluent 2020 R2 was used to analyze the flow field of the converter static steelmaking process. The aperture of the steel outlet and the timing of the vortex formation under different angles were studied, as well as the disturbance level of the injection flow in the ladle molten pool. The study revealed that in the steelmaking process, the emergence of tangential vectors caused the entrainment of slag by the vortex, whereas in the later stages of steelmaking, the turbulent flow of slag disrupted the vortex, resulting in its dissipation. When the converter angle increases to 90°, 95°, 100°, and 105°, the eddy current occurrence time is 43.55 s, 66.44 s, 68.80 s, and 72.30 s, and the eddy current stabilization time is 54.10 s, 70.36 s, 70.95 s, and 74.26 s, respectively. When the converter angle is 100–105°, it is suitable to add alloy particles into the ladle molten pool. When the tapping port diameter is 220 mm, the eddy current inside the converter changes and the mass flow rate of the tapping port is “oscillating”. When the aperture of the steel outlet was 210 mm, the steelmaking time could be shortened by about 6 s without affecting the internal flow field structure of the converter.

## 1. Introduction

With the development of science and technology, the demand for high-quality steel in the market is increasing, which poses higher requirements for metallurgical technology. Improving the cleanliness of steel by reducing the entry of inclusions and then improving the quality of steel is an important direction for the development of the steel industry [1,2]. During the steelmaking process in the converter, as the liquid level gradually decreases, a rapidly rotating free surface vortex is easily formed. Once the free surface vortex is formed, it is prone to roll over the slag. When the height of the steel liquid level is lower than the critical height of the vortex, the phenomenon of vortex-induced slagging will be very serious. This phenomenon occurs in the steelmaking process of the converter, ladle, and tundish. This will lead to a reduction in the yield of steel liquid, a shortening of the service life of the converter, an increase in the content of inclusions, blockage of the nozzle, and other problems [3,4], causing considerable losses to production efficiency [5,6,7]. Therefore, effectively delaying the occurrence of vortex-induced slagging is a problem that must be solved in the production of clean steel [2,8].

The main reason for the formation of eddy currents is due to the rotational motion of the fluid, where at least one of the three rotational angular velocity components of the fluid is not zero, meaning that the fluid particles exhibit microscopic horizontal rotation. In the actual fluid flow process, due to the viscosity of the fluid, rotation generally occurs. Since the formation of eddy currents during the steelmaking process in the converter is inevitable, it is necessary to have a detailed understanding of the mechanism of eddy current formation in order to mitigate the adverse effects of eddy currents on slag entrapment. So far, the study of eddy currents in steelmaking vessels has mainly focused on the ladle [9,10] and tundish [11,12,13]. Due to the non-visible process, high-temperature environment, and rotation of the converter during the steelmaking process, there is not yet a complete understanding of the flow phenomena in the converter during the steelmaking process. During the steelmaking process in a converter, once the molten steel enters the tapping hole, it will have a certain initial velocity, and the tapping speed at different tapping angles is completely different. At the end of the converter steelmaking process, alloying treatment needs to be carried out on the molten steel, and the influence of the distribution of the alloy by the molten steel flowing out at different converter angles is also different. Referring to the automatic tapping process of an enterprise, the angle changes in a “step-like” manner over time, so it is particularly important to determine the optimal tapping angle for adding alloying materials. Therefore, the timing of alloy addition needs to ensure that the tapping time is short, the amount of eddy current is small, and the turbulence intensity in the ladle is high enough. 

Some researchers have studied factors such as different tapping hole sizes and different steel heights but have not considered the effect of different converter tilts on the steelmaking process. Wang et al. [14] discussed the influence of initial liquid level height on the timing of slag coiling and built a quarter-converter model through numerical simulation. However, because the structure near the tapping port of the converter is more complex, different from the tapping port of the ladle, tundish, and other symmetric structures. Therefore, Pang et al. [15] built a complete model of the converter steel drawing process, but the model ignored the slag layer. Therefore, a full three-dimensional transient multiphase flow model of the converter steelmaking process was established on the basis of the previous studies. The eddy current phenomenon in the converter steelmaking process was studied, and the optimal time for adding alloy materials was determined. The mass flow rate of the wire outlet under these conditions will provide a basis for the study of the melting trajectory and dynamics of alloy particles in the converter steelmaking process in the future. By parametric analysis of the influence of different parameters on the vortex entrain slag, the correlation between the tapping angle of the converter, tapping hole aperture, and alloy adding time was established. Based on existing research, a fully three-dimensional, transient, multiphase flow model is established for mathematical simulation of the converter steelmaking process, studying the eddy current phenomena during converter steelmaking, and determining the optimal timing for adding alloying materials. The quality flow rate of the tapping hole under this condition will be used for the study of the melting trajectory and dynamics of alloy particles in future converter steelmaking processes. By parameterizing the analysis of the influence of different parameters on the eddy current entraining furnace slag, the correlation between the converter inclination angle, tapping hole aperture, and the timing of alloy addition is established.

## 2. Model Description and Geometric Model

### 2.1. Computational Model and Meshing

A simplified three-dimensional mathematical model based on the actual size of a 300 t steelmaking converter in a steel plant was established (Figure 1a). During the converter smelting process, a layer of slag floats on the liquid surface, and air fills the remaining space. Based on the characteristics of converter steelmaking, the following assumptions were made when establishing the mathematical model: (1) the experimental environment is relatively stable, and the disturbance caused by air flow on the liquid surface can be ignored; (2) the inner wall and connections of the converter container are smooth, and the frictional effect can be neglected; (3) the temperature change during the experiment is slight, and the influence of temperature on liquid density and viscosity can be ignored; (4) in the turbulent mixing zone, the flow satisfies the basic assumptions of the thin turbulent shear layer theory.

To simulate the converter steelmaking process accurately, a model was built using SolidWorks software 2021 at a 1:1 scale with respect to the actual model. The model includes four computational domains: the converter molten pool domain, the converter air domain, the ladle, and the air domain between the end of the tapping hole and the top of the ladle. The numerical scheme is based on pressure, transient, and three-dimensional calculations. The scheme uses the converter angle (90°, 95°, 100°, 105°) and the tapping hole diameter (200 mm, 210 mm, 220 mm) as variables. The non-structured tetrahedral mesh is created using Workbench, and the Multzone method is used to divide the mesh for the ladle domain. The complete geometric modeling and mesh division of the calculation domain are shown in Figure 2. The calculation mesh consists of 1,600,000 elements, with a maximum positive skewness of 0.823 and a minimum orthogonal quality of 0.177. To more accurately simulate the behavior of the slag–air interface and the steel liquid flow area, local mesh refinement technology was used, as shown in Figure 2c.

### 2.2. Governing Equations

The process of vortex formation in the steel tapping process of the converter can be ignored in the study without considering the effect of temperature, and the control equations used in the simulation of this process are mainly the continuity equation, momentum equation, turbulent kinetic energy equation, and volume equation.

(1) The multiphase VOF (volume of fluid) model can be expressed as follows:

Continuity equation:(1)$$\frac{\partial (uA_{x})}{\partial x}+\frac{\partial (vA_{y})}{\partial y}+\frac{\partial (wA_{z})}{\partial z}=0$$

Momentum equation:(2)$$\frac{\partial u}{\partial \tau }+\frac{1}{V_{F}}(uA_{x}\frac{\partial u}{\partial x}+vA_{y}\frac{\partial u}{\partial y}+wA_{z}\frac{\partial u}{\partial z})=\frac{1}{\rho }\frac{\partial p}{\partial x}+G_{x}+f_{x}$$
(3)$$\frac{\partial v}{\partial \tau }+\frac{1}{V_{F}}(uA_{x}\frac{\partial v}{\partial x}+vA_{y}\frac{\partial v}{\partial y}+wA_{z}\frac{\partial v}{\partial z})=\frac{1}{\rho }\frac{\partial p}{\partial y}+G_{y}+f_{y}$$
(4)$$\frac{\partial w}{\partial \tau }+\frac{1}{V_{F}}(uA_{x}\frac{\partial w}{\partial x}+vA_{y}\frac{\partial w}{\partial y}+wA_{z}\frac{\partial w}{\partial z})=\frac{1}{\rho }\frac{\partial p}{\partial z}+G_{z}+f_{z}$$
where $G_{x}$, $G_{y}$, $G_{z}$ are the gravitational acceleration in directions x, y, z, respectively, m/s^2^; $f_{x}$, $f_{y}$, $f_{z}$ is the viscous force in direction x, y, z; $V_{F}$ is the volume fraction that can flow; ρ is fluid density, kg/m^3^; p is the pressure acting on the fluid element; u, v, w is the component of velocity in direction x, y, z, m/s.

(2) The turbulence modeling is expressed through the standard k-ε model, In the turbulent mixing zone, the flow satisfies the basic assumption of thin, turbulent shear layer theory.

Turbulence kinetic energy equation:(5)$$\frac{\partial (\rho k)}{\partial t}+\frac{\partial (\rho ku_{i})}{\partial x_{i}}=\frac{\partial }{\partial x_{j}}[(\mu +\frac{u_{i}}{\sigma k})\frac{\partial k}{\partial x_{i}}]+G_{k}+G_{b}-\rho \varepsilon -Y_{M}+S_{k}$$

Rate of dissipation equation:(6)$$\frac{\partial (\rho \varepsilon )}{\partial t}+\frac{\partial (\rho \varepsilon u_{i})}{\partial x_{i}}=\frac{\partial }{\partial x_{j}}[(\mu +\frac{u_{i}}{\sigma_{\varepsilon }})\frac{\partial \varepsilon }{\partial x_{i}}]+C_{1\varepsilon }\frac{\varepsilon }{k}(G_{k}+C_{3\varepsilon }G_{b})+C_{2\varepsilon }\rho \frac{\varepsilon^{2}}{k}+S_{\varepsilon }$$
where ρ is the fluid density; t is time; $u_{i}$ is the fluid flow velocity in direction i; $\mu_{t}$ is the turbulent viscosity coefficient; μ is the turbulent dynamic viscosity; $x_{i}$ and $x_{j}$ denote the Cartesian coordinates in directions i and j, respectively; denotes the turbulent energy due to the mean velocity gradient; $G_{b}$ denotes the turbulent energy due to buoyancy; $Y_{M}$ denotes the contribution of pulsating expansion to the total dissipation rate in compressible turbulent flow $C_{1\varepsilon }$, $C_{2\varepsilon }$, $C_{3\varepsilon }$, $\sigma_{k}$, and $\sigma_{\varepsilon }$ are the constants of the k-ε model, which are 1.43, 1.92, 0.09, 1.0 and 1.3, respectively; $S_{k}$ and $S_{\varepsilon }$ are the user-defined turbulent energy phase and dissipation rate source phase, respectively.

(3) For each phase in the model, introduce a variable called unit phase volume fraction, such that the sum of the volume fractions of all phases in each control volume is equal to 1.

For phase i, the volume fraction equation is:(7)$$\frac{\partial a_{i}}{\partial \tau }+\vec{v}\nabla a_{i}=S_{a_{i}/\rho_{i}}$$

The volume fraction equation for the primary phase is:(8)$$\sum_{i=1}^{n}a_{i}=1$$

In the equation, $a_{i}$ represents the volume occupied by the fluid in the i fluid volume segment, with a value between 0 and 1.

### 2.3. Boundary Conditions and Numerical Details

The Fluent 20 R2 fluid simulation software under the ANSYS software package was used to perform numerical simulations of the steelmaking process. The physical and chemical properties of the steel and air are shown in Table 1. Since the simulation focuses on the late stage of steelmaking in the converter, 70 t of molten steel were preset in the ladle, and 178 t of liquid steel and 22.24 t of slag were in the converter. Based on the actual flow of fluids in the converter, the boundary conditions of the model were determined as follows: the boundary condition at the converter mouth adopts a pressure inlet, the edge of the air domain above the ladle is a pressure outlet, the inlet and outlet gauge pressure are set to 0 Pa, and the steel liquid flows out freely due to gravity. The gravity acceleration rate $g=9.81\;m/s^{2}$, and no-slip wall surface is avoided. The PISO (pressure implicit with splitting of operators) algorithm was used to couple the pressure and velocity terms. In order to obtain more accurate solutions, second-order upwind schemes were used for the discretization of convective terms. The calculation residual was set to 10^−3^, and iterative calculations were performed after initialization. Calculating the fluxes was carried out using the gradient of each variable evaluated at the cell centroid. The numerical convergence was attained when the sum of the residuals for the flow variables was less than 10^−3^.

### 2.4. Grid-Independency Study

To eliminate the influence of grid resolution on numerical results, a grid-dependency study was conducted to compare the effects of three different grid sizes on the time of slag entrainment by vortex. The results show that the error in the onset and stabilization time of vortex flux on the 160 W and 250 W grids is 0.14% and 0.89%, respectively. The error in the onset and stabilization time of vortex on the 80 W and 160 W grids is 11.2% and 7.8%, respectively. Table 2 summarizes the key parameters of the grid-dependency study. It can be clearly seen that the 160 W grid satisfies the computational requirements of the turbulence model used in this paper.

## 3. Results and Discussion

### 3.1. The Tapping Flow Field of the Converter at 90 Tilt Angles

An analysis was conducted on the steelmaking process of a 200 mm diameter tapping hole with a 90° tapping angle in the converter. Figure 3 shows the distribution diagrams of steel and slag along the vertical section of the converter at 20 s, 50 s, 70 s, and 125 s after tapping. From Figure 3a, it can be seen that at the beginning of steel tapping, the steel and slag are mutually insoluble and clearly stratified, and the flow of the steel inside the converter is relatively stable without the formation of vortex.

Fluent monitoring shows that the time when slag flows out of the taphole is 43 s, while the time when stable vortex flow appears is 50 s. In the process of steel tapping (Figure 3b), many small vortices disappear as the flow develops [16]. When the fluid in the converter descends to a certain height, a sustained vortex flow will form. The front end of the vortex develops in the direction of the taphole, and the slag is entrained into the taphole, resulting in the phenomenon of vortexing and slag coiling (Figure 3c). At the end of the steelmaking process (Figure 3d), a surface depression can be observed on the slag layer.

During the development of the vortex flow (Figure 3a,b), although the mass flow rate of the steel through the tapping hole decreased, the impact of the jet flow on the ladle increased gradually due to the increase in the depth of the molten pool. This caused greater disturbance to the steel inside the ladle. As the depth of the steel increases, the eddies and vortices generated by the impact of the jet flow will more strongly affect the gas inside the steel. However, if the depth of the steel bath in the ladle is large, the distance between the bottom of the ladle and the impact point of the nozzle will become very large, and the kinetic energy of the impact will be absorbed and dispersed by the steel liquid. At this time, the disturbance effect of the impact will decrease.

During the growth period of a vortex (Figure 3c,d), the amount of air drawn into the steel bath by the descending jet decreases gradually. The stirring effect produced in the steel bath also weakens, because the viscosity of slag is higher than that of steel, and the propagation speed of the jet impact in slag is also slower. According to Figure 4, due to the density difference between the slag and steel, the presence of slag forms a “buffer zone”, making it difficult for the impact of the jet flow to directly affect the steel, thereby reducing the disturbance to the steel [17]. As the thickness of the slag layer increases, the disturbance caused by the injection flow to the molten steel in the ladle gradually decreases.

### 3.2. Analysis of Vortex Formation during Converter Tapping

During the process of producing steel in a converter, the molten steel is continuously impacted and stirred against the furnace wall due to unstable and varying flow velocities, resulting in the formation of a rotating vortex. This vortex has both horizontal and vertical rotations. As the rotation process is highly complex, the formation of the converter steel vortex is initially discussed by analyzing the fixed tilt angle. The formation of the vortex is related to the speed and direction of the steel liquid flow. Above the taphole, the flow rate of the steel liquid is faster, and the steel liquid inside the converter needs to pass through a conical position to give the fluid an initial axial velocity. During the tapping process, the turbulence disturbance causes the tangential flow to become unstable and easily form eddies.

Figure 5 shows the velocity streamlines at different moments during the tapping process of the converter at an angle of 100°. The vortex initiation time is 68.8 s, and the stable development time is 70.95 s. As shown in Figure 5a, when the converter begins tapping, the static pressure of the steel liquid near the taphole is no longer balanced due to the effect of gravity. The pressure difference causes the steel liquid to flow out quickly, and the velocity streamlines are relatively uniform and neat. As shown in Figure 5b, although most of the flow directions of the steel liquid streamlines still point to the centerline of the taphole, some flow directions near the mouth begin to align with the tangential direction of the taphole, and there is no vortex above the taphole at this time. As the tapping process continues, two vortices appear above the taphole in the early stage of vortex initiation (Figure 5c), and the centers of the two vortices are connected by rotation in the appropriate tangential direction, causing the flow direction of the fluid streamlines to change, and countless micro-vortices develop into visible vortices. With the further extension of tapping time, stable vortices are formed, and the phenomenon of vortex slag coiling appears (Figure 5d). As shown in Figure 5e,f, after the vortex is stably formed, the vortex gradually becomes smaller and even disappears as the tapping process progresses. On the one hand, this is because the overall flow rate of the taphole decreases in the late stage of tapping. On the other hand, when the slag amount is too large, the flow velocity of the slag liquid will increase, which will intensify the turbulent motion of the slag liquid, destroying the formation and maintenance of the vortex and causing the number and size of the vortices to decrease.

It can be seen that when a tangential vector appears above the tundish nozzle, the steel liquid vortex begins to form. As the vortex develops, the turbulence of the steel liquid streamlines above the tundish nozzle increases, and the steel liquid vortex becomes stronger and gradually forms a whirlpool, which begins to roll into the slag, forming a phenomenon of swirling slag. With the increase in the slag amount during the steel pouring process, the turbulent motion of the slag liquid destroys the connection of the vortex, resulting in the gradual disappearance of the vortex.

### 3.3. Comparison of Flow Field with Different Converter Tilt Angles

When the converter tilt angle is fixed, the self-gravity of the fluid in the steel ladle forms a static pressure difference, which is the energy source for the development of vortex motion. Figure 6e–h shows the position enlarged velocity flow diagram of the red box in Figure 6a–d. The red line in Figure 6e–h is the slag layer. Due to the angle of 129° between the tapping hole and the left wall, and the angle of 100° between the tapping hole and the furnace wall, as the tilt angle of the converter increases, the vortex currents in the slag layer gradually shift from above the tapping hole to the right side, reducing the tangential velocity and disturbance of the slag layer. When the converter is tilted at an angle of 105° and the tapping time is 40 s, the angles on both sides of the tapping hole tend to be symmetric, and the tangential vector becomes very small (Figure 6). This suppresses the development of a slag layer vortex, even approaching a converging vortex [18]. In other words, when the tilt angle of the converter is 105°, the vortex in the slag layer has basically stopped rotating.

The monitoring standard for slag entrapment is to measure the flow rate of slag by monitoring the cross-section of the tapping hole during steelmaking. The onset times of vortex entrainment for converter angles of 90°, 95°, 100°, and 105° are 43.55 s, 66.44 s, 68.80 s, and 72.30 s, respectively. The stable development times of vortex entrainment are 54.10 s, 70.36 s, 70.95 s, and 74.26 s, respectively. Vortexes exist from the beginning of the steel flow, even if the steel–slag interface is not concave, but vortex motion has formed at the bottom. With an increase in converter angle, the tangential vector above the tapping hole tends to be symmetrical, and the duration of stable vortex entrainment decreases. As the initial tangential disturbance is the main factor for the vortex formation [19], the asymmetry of the tangential vector above the tapping hole leads to different onset and development times for the vortex.

According to Figure 7, when slag occurs at different angles of the converter, the remaining liquid steel in the converter is 5.12m^3^, 1.45m^3^, 0.80m^3^ and 0.54m^3^, respectively. When the tapping hole is close to the vertical ground, the angle of the converter has little effect on the timing of slagging, but it has a significant impact on the speed of the tapping hole. As the tapping hole is vertical to the ground, the static pressure above the tapping hole caused by gravity is the largest, and the steel liquid flow rate at an angle of 100° is the fastest. Due to the special nature of the converter’s tapping hole, the angles of the left and right walls of the tapping hole are inconsistent, so when the angle of the converter is 105°, the tapping speed is slightly reduced, and the timing of slagging is delayed.

When the converter angle is 100° (Figure 8c), the maximum steel pouring velocity is obtained. At this time, the pouring angle is perpendicular to the center of the bottom of the ladle, and the distribution of the velocity vector inside the ladle is nearly symmetrical. This is because the wall above the taphole of the converter is an asymmetric interface, which causes the velocity vector distribution in the pouring of molten steel to be incompletely symmetrical. When the converter angle is 105° (Figure 8d), the stagnant area of the molten pool vector inside the ladle is larger. The pouring velocities of molten steel are relatively close at converter angles of 100° and 105°. However, due to differences in the pouring location and angle deviation, the inner wall of the ladle on one side is eroded when the converter angle is 105°. This is because as the pouring angle of the molten steel increases, the number and size of the vortex will change. When the molten steel flows into the ladle at a larger angle from the nozzle, the eddies generated in the molten pool can more strongly disturb the molten pool, thereby increasing the turbulence level in the molten pool [20]. This helps to mix the chemical components in the molten pool, thus improving the uniformity and quality of the molten steel and reducing the temperature gradient in the molten pool.

In the late stage of steelmaking, alloy materials are added to the steelmaking furnace. The influencing factors of the movement trajectory and residence time of alloy particles in the molten steel include the physical and chemical parameters of the alloy particles, the steel flow field [21,22], and the location where the alloy particles are added to the steelmaking furnace [20,23,24]. In fact, the alloying process of alloy particles in the steelmaking furnace is mainly carried out by the mechanisms of diffusion and mixing. Diffusion refers to the free diffusion of alloy elements in the steel, and mixing refers to the process in which the turbulent action formed in the steel promotes the mixing of the steel and slag in the molten steel. Therefore, when the angle of the converter is between 100–105°, it is suitable to add alloy particles to the molten steel in the steelmaking furnace. At this time, the flow rate of the steel liquid is relatively high, and the critical height of the molten slag in the converter is relatively low.

### 3.4. Influence of Different Tapping Hole Sizes on the Flow Field

Figure 9 shows the mass flow rate of steel, slag, and molten steel velocity, as well as the residual volume of molten steel in the converter during the second tapping process with different tapping hole diameters (a) 200 mm, (b) 210 mm, and (c) 220 mm, for 50 s. The time when vortex currents start to roll slag is 68.80 s, 62.64 s, and 57.11 s, respectively. The time when vortex currents start to develop stably is 70.95 s, 64.49 s, and 58.00 s, respectively. As the diameter of the tapping hole increases, the time for the vortex to reach the tapping hole shows a decreasing trend. The residual volume of molten steel in the converter is 0.80 m^3^, 0.77 m^3^, and 0.73 m^3^, respectively, when the different tapping hole diameters start to roll slag, indicating that the critical height of the vortex decreases with an increase in the tapping hole diameter. This is because, during the tapping process, the effect of the tapping hole diameter on the initiation of the vortex is much smaller than the effect of the mass flow rate of the molten steel from the tapping hole. With a larger tapping hole diameter, the mass flow rate of molten steel increases, and the liquid level drops faster. Although the critical height of the liquid level for vortex formation is lower, the development time of the vortex accelerates, and the time for the vortex to reach the tapping hole becomes shorter.

However, overall, the effect of increasing the tapping hole diameter on the formation of the vortex is not significant. However, there is a phenomenon where periodic oscillations occur in the flow of molten iron from the taphole when the taphole diameter is increased to 220 mm, which disappears only at the end of tapping. This “oscillation” phenomenon is due to the dynamic instability of the steel in the converter. The occurrence of oscillation tapping can cause fluctuations in the quality of the steel in the furnace, thereby affecting the stability and efficiency of the steel production process. Increasing the taphole diameter of the converter will increase the tapping speed, reduce the surface tension of the molten steel, and reduce the viscosity of the molten steel.

When the diameter of the tapping hole is small, the flow of molten steel is mainly restricted by the hole size, and the flow rate is relatively stable, so it is not easy to cause oscillation. However, when the diameter of the tapping hole increases, the degree of restriction on the flow of molten steel decreases, and the instability of the flow increases, making it easy to cause oscillation. The fluid flow inside the furnace and the velocity and mass flow rate at the tapping hole are closely related. As shown in Figure 10, when the diameter of the tapping hole increases from 200 mm to 220 mm, the symmetric vector position in the converter is closer to the top of the tapping hole. When the diameter of the tapping hole is 220 mm, the internal flow structure of the converter changes after 50 s of flow (Figure 10c), which produces a large vortex, further causing the flow of molten steel to oscillate. When the flow velocity at the tapping hole increases, the kinetic energy of the molten steel also increases, and therefore the mass flow rate also increases accordingly. Figure 10g shows clearly the counteractive effect of two vortices [25]. When the flows meet, the counteractive effect reduces the tangential velocity between the top of the tapping hole, suppresses the formation of the free surface vortex, and reduces the height of the adjacent vortex slag. Therefore, a diameter of 210 mm for the steel tapping hole is the optimal choice, as it can reduce the tapping time of the converter without changing the internal flow field structure of the converter.

## 4. Conclusions

This paper uses the VOF multiphase flow model and combines it with the 300 t converter steelmaking process to establish a slag entrainment by vortex Flux model for the converter steelmaking process. By discussing the converter steelmaking process at different converter angles and different steel tapping orifice diameters, the following conclusions can be drawn:

1. As the steel pouring process continues, the vortex develops into a whirlpool due to the appearance of tangential vectors above the converter taphole, and the phenomenon of swirling slag occurs. At the end of the steel pouring, as the amount of slag increases, the turbulent motion of the slag liquid disrupts the connection of the vortex, resulting in the gradual disappearance of the vortex.

2. When the converter angle is 90°, 95°, 100°, and 105°, the vortex slagging times are 43.55 s, 66.44 s, 68.80 s, and 72.30 s, respectively. As the converter angle increases, the time for forming a stable vortex becomes shorter. When the converter angle is between 100–105°, the internal flow field of the steelmaking furnace is more suitable for adding alloy particles.

3. When the steel tapping orifice diameter is 200 mm, 210 mm, and 220 mm, the steel liquid residue starts to swirl at 0.80 m^3^, 0.77 m^3^, and 0.73 m^3^, respectively. The initiation times of vortex slag were 68.80 s, 62.64 s, and 57.11 s, respectively. The stable development time of vortex slag was 70.95 s, 64.49 s, and 58.00 s, respectively. When the steel tapping orifice diameter is 210 mm, the steel tapping speed can be increased without affecting the internal flow field structure of the converter. When the steel tapping orifice diameter is 220 mm, the mass flow rate of the steel tapping orifice shows an “oscillating” phenomenon.

## Figures and Tables

**Figure 1 materials-16-03209-f001:**
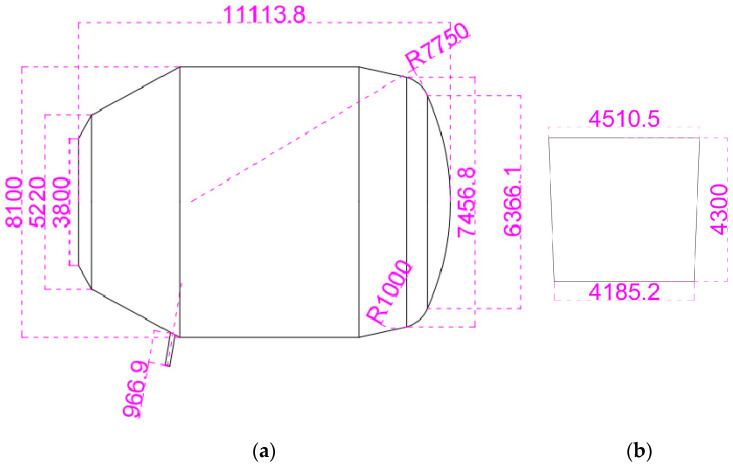
Geometric of the (**a**) converter; (**b**) ladle.

**Figure 2 materials-16-03209-f002:**
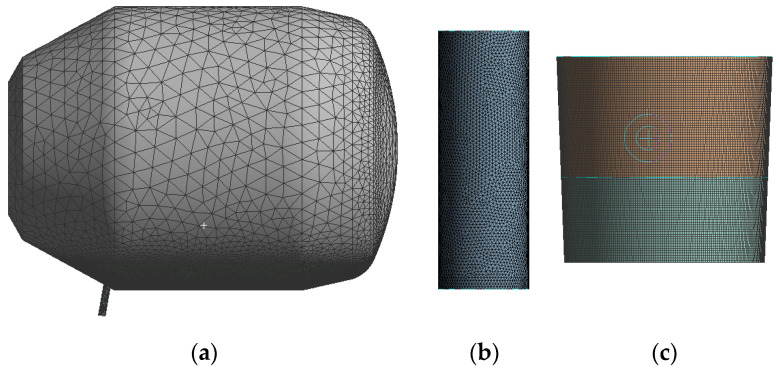
Diagram meshing of (**a**) converter; (**b**) surround; (**c**) ladle.

**Figure 3 materials-16-03209-f003:**
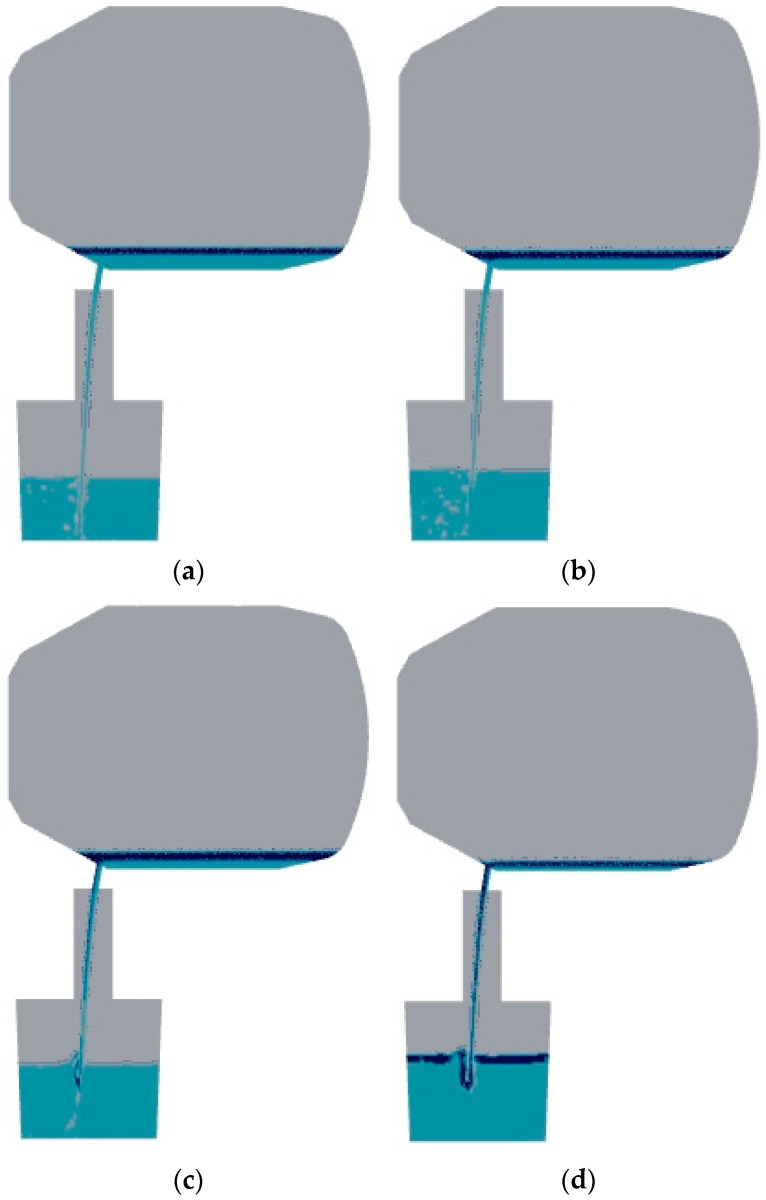
Steel distribution diagram during the 90° tapping process at different times. (**a**) 20 s; (**b**) 50 s; (**c**) 70 s; (**d**) 125 s.

**Figure 4 materials-16-03209-f004:**
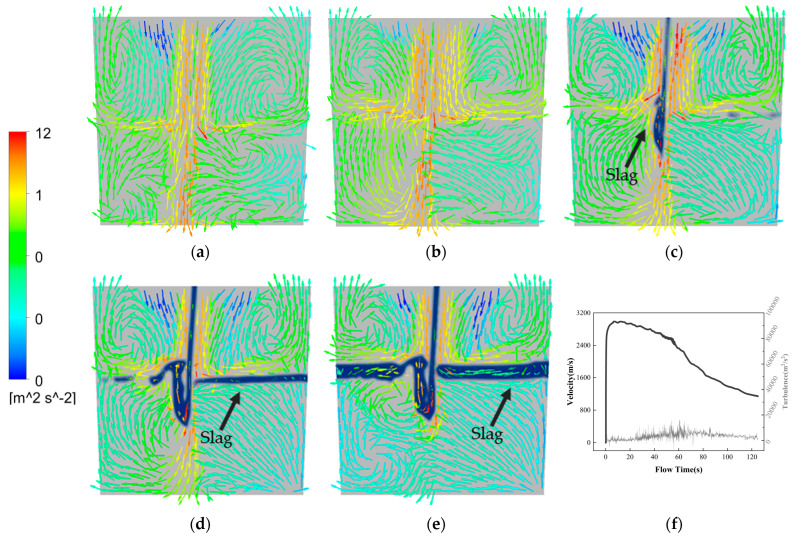
Turbulent flow distribution diagram in the ladle at (**a**) 20 s; (**b**) 50 s; (**c**) 70 s; (**d**) 90 s; (**e**) 125 s during steel tapping when the converter is at 90° and (**f**) velocity vector and turbulence intensity distribution.

**Figure 5 materials-16-03209-f005:**
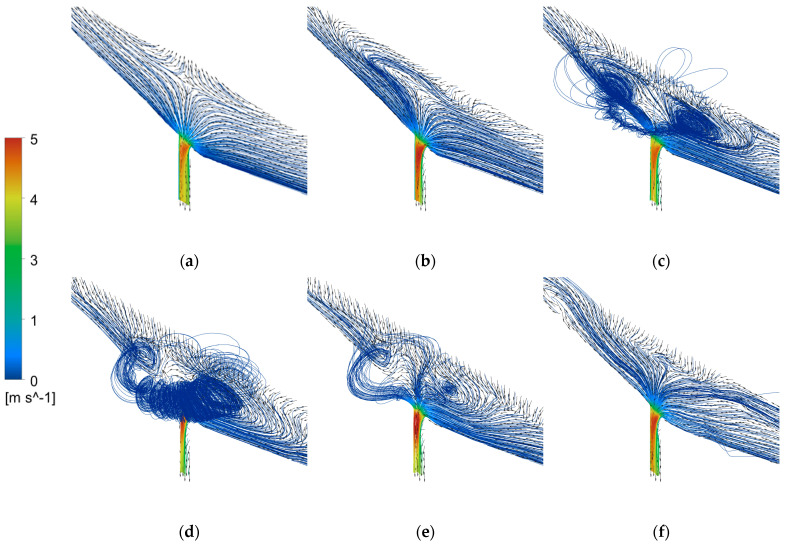
Velocity streamline diagram at a converter angle of 100°: (**a**) 1 s; (**b**) 30 s; (**c**) 67 s; (**d**) 72 s; (**e**) 75 s and (**f**) 80 s.

**Figure 6 materials-16-03209-f006:**
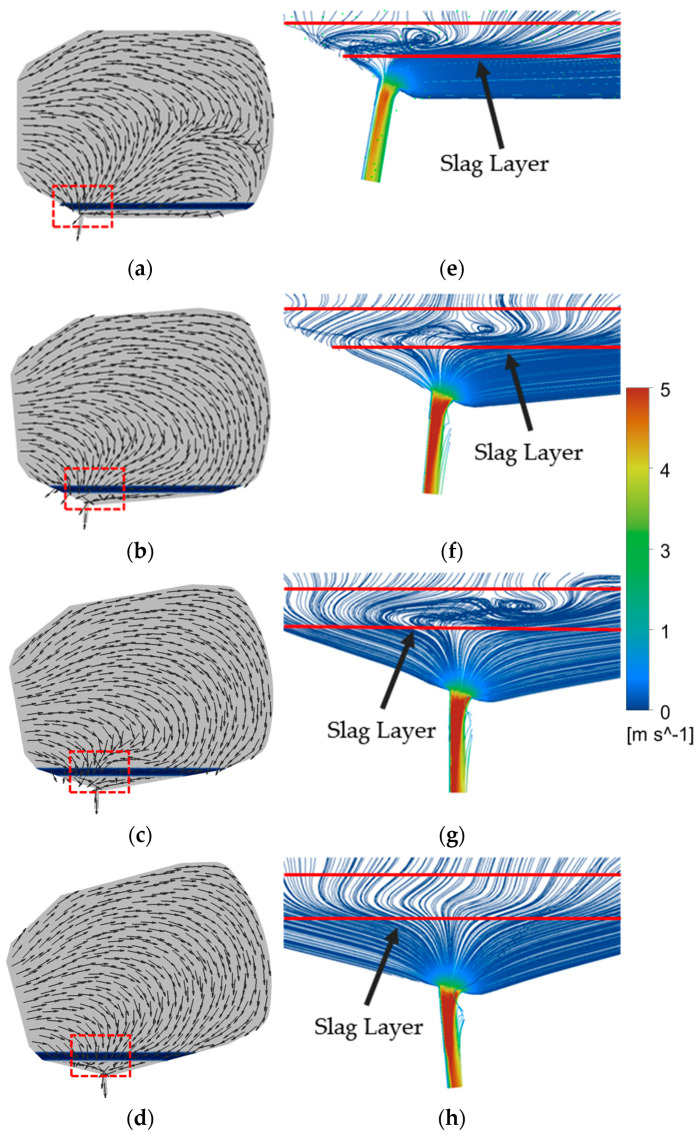
Velocity vector diagram at converter angles (**a**) 90°, (**b**) 95°, (**c**) 100°, (**d**) 105°, and streamlines diagram at angles (**e**) 90°, (**f**) 95°, (**g**) 100°, (**h**) 105°.

**Figure 7 materials-16-03209-f007:**
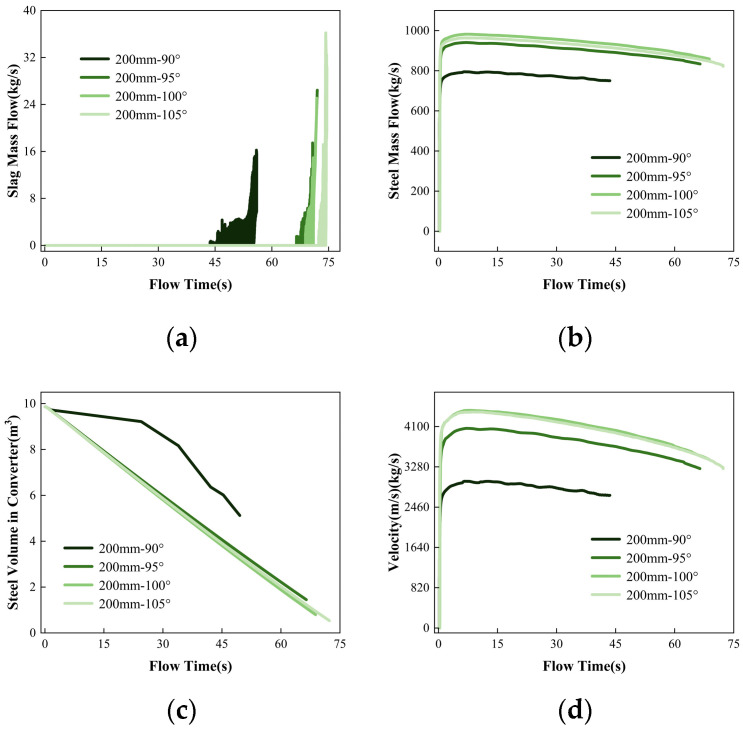
The variation trend of (**a**) slag mass flow; (**b**) steel mass flow; (**c**) steel volume in converter and (**d**) tapping velocity with flowing time when the tilting angle of the converter is 90°, 95°, 100°, or 105°.

**Figure 8 materials-16-03209-f008:**
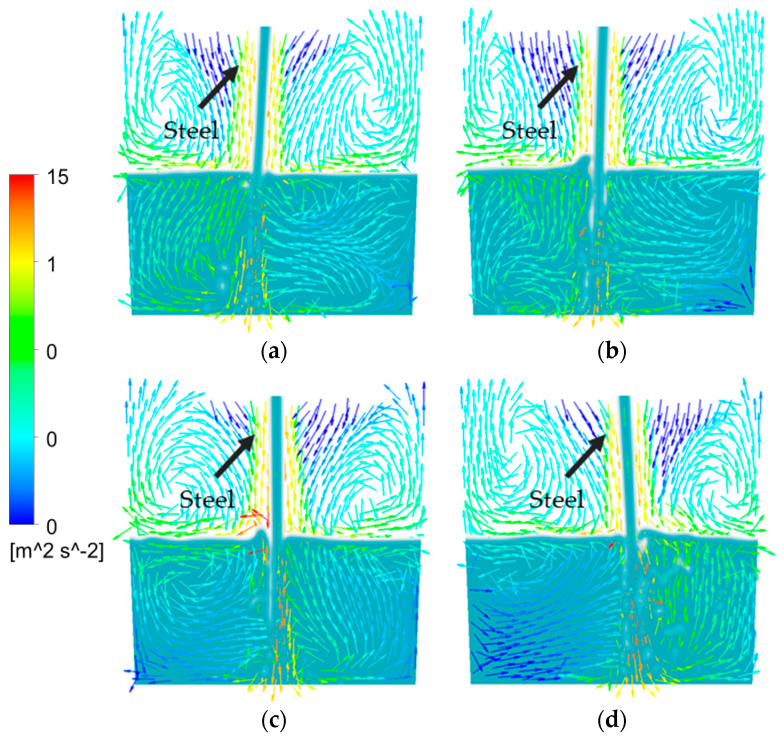
Velocity vector diagram of the ladle at different tapping angles of the converter: (**a**) 90°; (**b**) 95°; (**c**) 100°; (**d**) 105°.

**Figure 9 materials-16-03209-f009:**
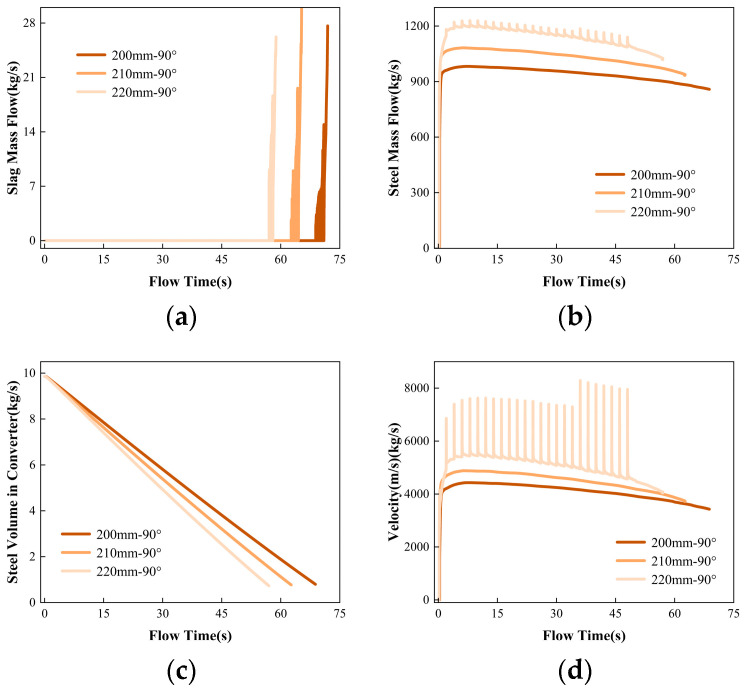
The variation trend of (**a**) slag mass flow; (**b**) steel mass flow; (**c**) steel volume in converter, and (**d**) tapping velocity with flowing time when the tilting hole sizes of the converter are 200 mm, 210 mm, or 220 mm.

**Figure 10 materials-16-03209-f010:**
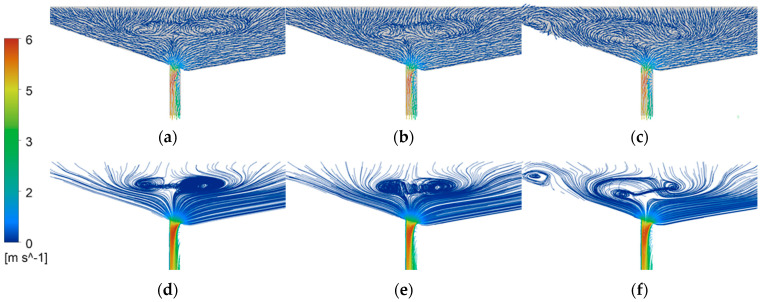
Velocity vector distribution of converter tapping hole with diameters of (**a**) 200 mm, (**b**) 210 mm, and (**c**) 220 mm, as well as (**d**) 200 mm, (**e**) 210 mm, and (**f**) 220 mm velocity streamlines distribution.

**Table 1 materials-16-03209-t001:** Physical parameters of fluid used in simulation.

Property	Air	Steel	Slag
Density, kg/m^3^	1.225	7100	2700
Viscosity, pa·s	1.79 × 10^−5^	0.0065	0.1998

**Table 2 materials-16-03209-t002:** Grid dependency study based on the case.

Number of Grid	80 w	160 w	250 w
Vortex onset time	38.68	43.55	43.49
Vortex stabilization time	49.84	54.10	54.58

## Data Availability

Not applicable.

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
