# Peer review of "Numerical Simulation of Slag Entrainment by Vortex Flux during Tapping at Converter"

_materials, 2023, doi:10.3390/ma16083209_

Round 1

Reviewer 1 Report

·       The article was written well and has good technological material.

·       A decent amount of literature survey has been carried-out.

·       The results were discussed and explained with Figures,  and References.

·       The results and discussion part should be improved with few latest references.

·       Beside the technical comments that must be addressed, as noted in the critiques, the technical English grammar requires improvement.

  • There are many typo mistakes - requires improvement.
  • The obtained results are in line with the literature.
  • Modelling details need to be presented and discussed.
  • The details of the convertor need to be presented.
  • Composition of the steel alloy need to be discussed and the effect of alloying elements and temperature on the flow characteristics need to discussed in detail.
  • Justification for the selection of multiphase VOF model
  • The obtained results may be compared with the physically obtained results and the results need to be validated.

Reviewer 2 Report

The authors presented an article about “Numerical Simulation of Slag Entrainment by Vortex Flux During Tapping at Converter.” The authors have carried out a useful study on the improvement of steel casting. The article study is on numerical analysis. I can say that I like the numerical work and congratulate the authors in this respect. However, since the authors did not support these simulated results with experimental data, the reliability of the results cannot be fully established. This can be said as a serious shortcoming for the authors.But the article overall looks good. I think the paper is well organized and appropriate for the “Materials” journal, but the paper will be ready for publication after major revision.

·       The abstract looks good. Please include all significant results

·       The introduction part is very short. In addition, the number of citations made to 2019 and beyond is very few. Please benefit from more up-to-date articles in the intorduction section.

·       The aim of the article, its difference from other studies and the solutions it will bring to industrial problems should be stated in the last paragraph in the introduction part. In addition, short information should be given in the last paragraph with the article layout.

·       The mesh independent test, which shows the independence of the results from the mesh, should be performed and stated in the paper. Please explain y+ value.

·       Please specify the dimensions of the geometric model.

·       Have the authors tried different iteration numbers? Why is the number of iterations set to 50? Please explain in detail in the paper.

·       How do the authors explain the accuracy of the simulation model? Please explain in the paper.

·       Velocity vector diagrams and streamlines diagrams are too small. Please increase your visibility.

·       Discussion of the findings obtained in the article is somewhat lacking. Please comment further by comparing the results obtained.

Reviewer 3 Report

The study was conducted to improve the yield and quality of steel produced in the converter during the steelmaking process. CFD fluid simulation software Fluent 2020 R2 was used to analyze the flow field in the converter and ladle during the steelmaking process. The study focused on the aperture of the steel outlet, timing of vortex formation under different angles, and disturbance level of injection flow in the ladle molten pool.

The results of the study revealed that the emergence of tangential vectors caused the entrainment of slag by the vortex during steelmaking, whereas the turbulent flow of slag disrupted the vortex in the later stages, leading to its dissipation. It was found that the time from vortex formation to stability decreased as the angle of the converter increased. When the converter angle was between 100° - 105°, it was suitable to add alloy particles to the ladle molten pool. Additionally, the steelmaking time could be shortened by about 6 s without affecting the internal flow field structure of the converter when the aperture of the steel outlet was 210 mm.

Some relevant applications of the study to the production of novel alloyed structural materials for heavy duty tribo-fatigue systems could be referred to in the paper:

(i) on the development of mechanothermodynamics as a new branch of physics; (ii) research on tensile behaviour of new structural material monica; (iii) from fatigue and tribology to tribo-fatigue; (iv) on the development of tribo-fatigue as the new section of mechanics; (v) comparison of stress-strain states of rail-wheel pair made of steel and monica.

VOF abbreviation at page 3 should be disclosed.

Displacements u,v,w and time tau introduced in (1)-(4) should be explained after these formulas.

The phrase “From Figure 2(a), it can be seen that at the beginning of steel tapping, the steel and slag are completely mixed and clearly stratified” at page 5 is not clear since “completely mixed” and “clearly stratified” seem to be contradictory.

Figure 3 f should be improved or displayed as a separate one with clear definition of velocity and turbulence.

At page 6 it is said that “The vortex initiation time is 68.8 s”, however figure 4 c shows that vortex occurred at 67s.

Tapping angle for which the analysis of tapping diameter is made should be indicated in section 3.4.

The paper “Numerical Simulation of Slag Entrainment by Vortex Flux During Tapping at Converter” may be considered for publication in Materials after addressing the above comments.

Round 2

Reviewer 2 Report

Thank you for reply